# Universal Rate-Distortion-Perception Representations for Lossy Compression

**George Zhang**
Electrical and Computer Engineering
University of Toronto
gq.zhang@mail.utoronto.ca

**Jingjing Qian**
Electrical and Computer Engineering
McMaster University
qianj40@mcmaster.ca

**Jun Chen**
Electrical and Computer Engineering
McMaster University
chenjun@mcmaster.ca

**Ashish Khisti**
Electrical and Computer Engineering
University of Toronto
akhisti@ece.utoronto.ca

## Abstract

In the context of lossy compression, Blau & Michaeli [5] adopt a mathematical notion of perceptual quality and define the information rate-distortion-perception function, generalizing the classical rate-distortion tradeoff. We consider the notion of universal representations in which one may fix an encoder and vary the decoder to achieve any point within a collection of distortion and perception constraints. We prove that the corresponding information-theoretic universal rate-distortion-perception function is operationally achievable in an approximate sense. Under MSE distortion, we show that the entire distortion-perception tradeoff of a Gaussian source can be achieved by a single encoder of the same rate asymptotically. We then characterize the achievable distortion-perception region for a fixed representation in the case of arbitrary distributions, and identify conditions under which the aforementioned results continue to hold approximately. This motivates the study of practical constructions that are approximately universal across the RDP tradeoff, thereby alleviating the need to design a new encoder for each objective. We provide experimental results on MNIST and SVHN suggesting that on image compression tasks, the operational tradeoffs achieved by machine learning models with a fixed encoder suffer only a small penalty when compared to their variable encoder counterparts.

## 1   Introduction

Unlike in lossless compression, the decoder in a lossy compression system has flexibility in how to reconstruct the source. Conventionally, some measure of distortion such as mean squared error, PSNR or SSIM/MS-SSIM [36, 37] is used as a quality measure. Accordingly, lossy compression algorithms are analyzed through rate-distortion theory, wherein the objective is to minimize the amount of distortion for a given rate. However, it has been observed that low distortion is not necessarily synonymous with high perceptual quality; indeed, deep learning based image compression has inspired works in which authors have noted that increased perceptual quality may come at the cost of increased distortion [1, 4]. This culminated in the work of Blau & Michaeli [5] who propose the rate-distortion-perception theoretical framework.

The main idea was to introduce a third *perception* axis which more closely mimics what humans would deem to be visually pleasing. Unlike distortion, judgement of perceptual quality is taken to

be inherently no-reference. The mathematical proxy for perceptual quality then comes in the form of a divergence between the source and the reconstruction *distributions*, motivated by the idea that perfect perceptual quality is achieved when they are identical. Leveraging generative adversarial networks [11] in the training procedure has made such a task possible for complex data-driven settings with efficacy even at very low rates [33]. Naturally, this induces a tradeoff between optimizing for perceptual quality and optimizing for distortion. But in designing a lossy compression system, one may wonder where exactly this tradeoff lies: is the objective tightly coupled with optimizing the representations generated by the encoder, or can most of this tradeoff be achieved by simply changing the decoding scheme?

Our contributions are as follows. We define the notion of *universal* representations which are generated by a fixed encoding scheme for the purpose of operating at multiple perception-distortion tradeoff points attained by varying the decoder. We then prove a coding theorem establishing the relationship between this operational definition and an information universal rate-distortion-perception function. Under MSE distortion loss, we study this function for the special case of the Gaussian distribution and show that the penalty in fixing the representation map with fixed rate can be small in many interesting regimes. For general distributions, we characterize the achievable distortion-perception region with respect to an arbitrary representation and establish a certain approximate universality property.

We then turn to study how the operational tradeoffs achieved by machine learning models on image compression under a fixed encoder compared to varying encoders. Our results suggest that there is not much loss in reusing encoders trained for a specific point on the distortion-perception tradeoff across other points. The practical implication of this is to reduce the number of models to be trained within deep-learning enhanced compression systems. Building on [30, 31], one of the key steps in our techniques is the assumption of *common randomness* between the sender and receiver which will turn out to reduce the coding cost. Throughout this work, we focus on the scenario where a rate is fixed in advance. We address the scenario when the rate is changed in the supplementary.

## 2 Related Works

Image quality measures include full-reference metrics (which require a ground truth as reference), or no-reference metrics (which only use statistical features of inputs). Common full-reference metrics include MSE, SSIM/MS-SSIM [36, 37], PSNR or deep feature based distances [16, 41]. No-reference metrics include BRISQUE/NIQE/PIQE [23, 24, 35] and Fréchet Inception Distance [14]. Roughly speaking, one can consider the former set to be distortion measures and the latter set to be perception measures in the rate-distortion-perception framework. Since GANs capable of synthesizing highly realistic samples have emerged, using trained discriminators as a proxy for perceptual quality in deep learning based systems has also been explored [17]. This idea is principled as various GAN objectives can be interpreted as estimating particular statistical distances [3, 25, 26].

Rate-distortion theory has long served as a theoretical foundation for lossy compression [7]. Within machine learning, variations of rate-distortion theory have been introduced to address representation learning [2, 6, 34], wherein a central task is to extract useful information from data on some sort of budget, and also in the related field of generative modelling [15]. On the other hand, distribution-preserving lossy compression problems have also been studied in classical information theory literature [27, 28, 40].

More recently, in an effort to reduce blurriness and other artifacts, machine learning research in lossy compression has attempted to incorporate GAN regularization into compressive autoencoders [1, 5, 33], which were traditionally optimized only for distortion loss [21, 32]. This has led to highly successful data-driven models operating at very low rates, even for high-resolution images [22]. An earlier work of Blau & Michaeli [4] studied only the perception-distortion tradeoff within deep learning enhanced image restoration using GANs. This idea was then incorporated with distribution-preserving lossy compression [33] to study the rate-distortion-perception tradeoff in full generality [5].

The work most similar to ours is [39], who observe that an optimal encoder for the "classic" rate-distortion function is also optimal for perfect perceptual compression at twice the distortion. Our work investigates the intermediate regime and also includes common randomness as a modelling

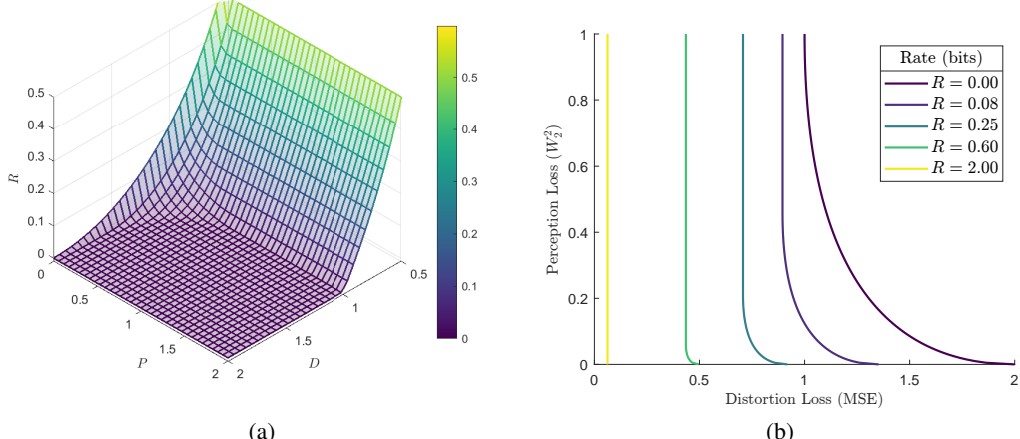

(a)    (b)

Figure 1: (a) The information Rate-Distortion-Perception function $R(D, P)$ for a standard Gaussian source $X$. (b) Distortion-perception cross-sections across multiple rates. The tension between perception and distortion is most visible at low rates. When both $P$ and $D$ are active, Theorem 3 implies that the rate needed to achieve an entire cross-section along fixed rate is the same as the rate to achieve any single point on the cross-section in the asymptotic setting.

assumption, which in principle allows us to achieve perfect perceptual quality at lower than twice the distortion. The concurrent work [10] also establishes the achievable distortion-perception region as in our Theorem 4 and provides a geometric interpretation of the optimal interpolator in Wasserstein space.

## 3    Rate-Distortion-Perception Representations

The backbone of rate-distortion theory characterizes an (operational) objective expressing what can be achieved by encoders and decoders with a quantization bottleneck in terms of an *information* function which is more convenient to analyze. Let $X \sim p_X$ be an information source to be compressed through quantization. The quality of the compressed source is measured by a distortion function $\Delta : \mathcal{X} \times \mathcal{X} \to \mathbb{R}_{\geq 0}$ satisfying $\Delta(x, \hat{x}) = 0$ if and only if $x = \hat{x}$. We distinguish between the *one-shot* scenario in which we compress one symbol at a time, and the *asymptotic* scenario in which we encode $n$ i.i.d. samples from $X$ jointly and analyze the behaviour as $n \to \infty$. The minimum rate needed to meet the distortion constraint $D$ on average is denoted by $R^*(D)$ in the one-shot setting and by $R^{(\infty)}(D)$ in the asymptotic setting. These are studied through the information rate-distortion function

$$R(D) = \inf_{p_{\hat{X}|X}} I(X; \hat{X}) \quad \text{s.t.} \quad \mathbb{E}[\Delta(X, \hat{X})] \leq D, \tag{1}$$

where $I(X; \hat{X})$ is the mutual information between a source $X$ and reconstruction $\hat{X}$. The principal result of rate-distortion theory states that $R^{(\infty)}(D) = R(D)$ [7]. Furthermore, it is also possible to characterize $R^*(D)$ using $R(D)$ as we will soon see.

In light of the discussion on perceptual quality, the flexibility in distortion function is not necessarily a good method to capture how realistic the output may be perceived. To resolve this, Blau & Michaeli [5] introduce an additional constraint to match the distributions of $X$ and $\hat{X}$ in the form of a non-negative divergence between probability measures $d(\cdot, \cdot)$ satisfying $d(p, q) = 0$ if and only if $p = q$. The one-shot rate-distortion-perception function $R^*(D, P)$ and asymptotic rate-distortion-perception function $R^{(\infty)}(D, P)$ are defined in the same fashion as their rate-distortion counterparts, which we will later make precise.

**Definition 1** (iRDPF). The information rate-distortion-perception function for a source $X$ is defined as

$$R(D, P) = \inf_{p_{\hat{X}|X}} I(X; \hat{X})$$

$$\text{s.t.} \quad \mathbb{E}[\Delta(X, \hat{X})] \leq D, \quad d(p_X, p_{\hat{X}}) \leq P.$$

The strong functional representation lemma [19, 31] establishes relationships between the operational and information functions:

$$R^{(\infty)}(D, P) = R(D, P), \tag{2}$$

$$R(D, P) \leq R^*(D, P) \leq R(D, P) + \log(R(D, P) + 1) + 5. \tag{3}$$

These results hold also for $R(D) = R(D, \infty)$. We make note that they were developed under a more general set of constraints for which stochastic encoders and decoders with a shared source of randomness were used. In practice, the sender and receiver agree on a random seed beforehand to emulate this behaviour.

## 3.1 Gaussian Case

We now present the closed form expression of $R(D, P)$ for a Gaussian source under MSE distortion and squared Wasserstein-2 perception losses (see also Figure 1(a) and Figure 1(b)). Recall that the squared Wasserstein-2 distance is defined as

$$W_2^2(p_X, p_{\hat{X}}) = \inf \mathbb{E}[\|X - \hat{X}\|^2], \tag{4}$$

where the infimum is over all joint distributions of $(X, \hat{X})$ with marginals $p_X$ and $p_{\hat{X}}$. Let $\mu_X = \mathbb{E}[X]$ and $\sigma_X^2 = \mathbb{E}[\|X - \mu_X\|^2]$.

**Theorem 1.** *For a scalar Gaussian source $X \sim \mathcal{N}(\mu_X, \sigma_X^2)$, the information rate-distortion-perception function under squared error distortion and squared Wasserstein-2 distance is attained by some $\hat{X}$ jointly Gaussian with $X$ and is given by*

$$R(D, P) = \begin{cases} \frac{1}{2} \log \dfrac{\sigma_X^2 (\sigma_X - \sqrt{P})^2}{\sigma_X^2 (\sigma_X - \sqrt{P})^2 - (\frac{\sigma_X^2 + (\sigma_X - \sqrt{P})^2 - D}{2})^2} & \\ \qquad\qquad\qquad if \ \sqrt{P} \leq \sigma_X - \sqrt{|\sigma_X^2 - D|}, \\ \max\{\frac{1}{2} \log \frac{\sigma_X^2}{D}, 0\} & if \ \sqrt{P} > \sigma_X - \sqrt{|\sigma_X^2 - D|}. \end{cases}$$

When $\sqrt{P} > \sigma_X - \sqrt{|\sigma_X^2 - D|}$, the perception constraint is inactive and $R(D, P) = R(D)$. The choice of $W_2^2(\cdot, \cdot)$ perception loss turns out to not be essential; we show in the supplementary that $R(D, P)$ can also be expressed under the KL-divergence.

## 3.2 Universal Representations

Whereas the RDP function is regarded as the minimal rate for which we can vary an encoder-decoder pair to meet any distortion and perception constraints $(D, P)$, the universal RDP (uRDP) function generalizes this to the case where we fix an encoder and allow only the decoder to adapt in order to meet multiple constraints $(D, P) \in \Theta$. For example, one case of interest is when $\Theta$ is the set of all $(D, P)$ pairs associated with a given rate along the iRDP function; how much additional rate is needed if this is to be achieved by a fixed encoder, rather than varying it across each objective? The hope is that the rate to use some fixed encoder across this set is not much larger than the rate to achieve any single point. As we will see for the Gaussian distribution, this is in fact the case in the asymptotic setting, and also approximately true in the one-shot setting. Below, we define the one-shot universal rate-distortion-perception function and the information universal rate-distortion-perception function, then establish a relationship between the two. In these definitions we assume $X$ is a random variable and $\Theta$ is an arbitrary non-empty set of $(D, P)$ pairs.

**Definition 2** (ouRDPF). A $\Theta$-universal encoder of rate $R$ is said to exist if we can find random variable $U$, encoding function $f_U : \mathcal{X} \to \mathcal{C}_U$ and decoding functions $g_{U,D,P} : \mathcal{C}_U \to \hat{\mathcal{X}}, (D, P) \in \Theta$ such that

$$\mathbb{E}[\ell(f_U(X))] \leq R, \quad \mathbb{E}[\Delta(X, \hat{X}_{D,P})] \leq D, \quad d(p_X, p_{\hat{X}_{D,P}}) \leq P,$$

where $\mathcal{C}_U$ is a uniquely decodable binary code specified by $U$, $\hat{X}_{D,P} = g_{U,D,P}(f_U(X))$, and $\ell(f_U(X))$ denotes the length of binary codeword $f_U(X)$. The random variable $U$ acts as a shared source of randomness. The infimum of such $R$ is called the one-shot universal rate-distortion-perception function (ouRDPF) and denoted by $R^*(\Theta)$. When $\Theta = \{(D, P)\}$, this specializes to the one-shot rate-distortion-perception function $R^*(D, P)$.

**Definition 3** (iuRDPF). Let $Z$ be a representation of $X$ (i.e. generated by some random transform $p_{Z|X}$). Let $\mathcal{P}_{Z|X}(\Theta)$ be the set of transforms $p_{Z|X}$ such that for each $(D, P) \in \Theta$, there exists $p_{\hat{X}_{D,P}|Z}$ for which

$$\mathbb{E}[\Delta(X, \hat{X}_{D,P})] \leq D \text{ and } d(p_X, p_{\hat{X}_{D,P}}) \leq P,$$

where $X \leftrightarrow Z \leftrightarrow \hat{X}_{D,P}$ are assumed to form a Markov chain. Define

$$R(\Theta) = \inf_{p_{Z|X} \in \mathcal{P}_{Z|X}(\Theta)} I(X; Z). \tag{5}$$

We refer to this as the information universal rate-distortion-perception function (iuRDPF) and say that the random variable $Z$ is a representation which is $\Theta$-universal with respect to $X$. The conditional distributions $p_{\hat{X}_{D,P}|Z}$ induce stochastic mappings transforming the representations to reconstructions $\hat{X}_{D,P}$ in order to meet specific $(D, P)$ constraints.

Note that we assume a shared source of stochasticity within the ouRDPF as a tool to prove the achievability of the iuRDPF, but not within the definition of the iuRDPF itself. Moreover, source $X$, reconstruction $\hat{X}_{D,P}$, representation $Z$, and random seed $U$ are all allowed to be multivariate random variables.

**Theorem 2.** $R(\Theta) \leq R^*(\Theta) \leq R(\Theta) + \log(R(\Theta) + 1) + 5.$

In practice, the overhead $\log(R(\Theta) + 1) + 5$ either makes the upper bound an overestimate of $R^*(\Theta)$ or is negligible compared to $R(\Theta)$. This overhead vanishes completely in the asymptotic setting as we will show in the supplementary. We can therefore interpret $R(\Theta)$ as the rate required to meet an entire set $\Theta$ of constraints with the encoder fixed. Within the set $\Theta$, it is clear that $\sup_{(D,P) \in \Theta} R(D, P)$ characterizes the rate required to meet the most demanding constraint. Now define

$$A(\Theta) = R(\Theta) - \sup_{(D,P) \in \Theta} R(D, P), \tag{6}$$

which is the rate penalty incurred by meeting *all* constraints in $\Theta$ with the encoder fixed. Let $\Omega(R) = \{(D, P) : R(D, P) \leq R\}$. It is ideal to have $A(\Omega(R)) = 0$ for each $R$ so that achieving the entire tradeoff with a single encoder is essentially no more expensive than to achieve any single point on the tradeoff, thereby alleviating the need to design a host of encoders for different distortion-perception objectives with respect to the same rate.

One can also take the following alternative perspective. The proof of Theorem 2 shows that every representation $Z$ can be generated from source $X$ using an encoder of rate $I(X; Z) + o(I(X; Z))$, and based on $Z$, the decoder can produce reconstruction $\hat{X}_{D,P}$ by leveraging random seed $U$ to simulate conditional distribution $p_{\hat{X}_{D,P}|Z}$. Therefore, the problem of designing an encoder boils down to identifying a suitable representation. Given a representation $Z$ of $X$, we define the achievable distortion-perception region $\Omega(p_{Z|X})$ as the set of all $(D, P)$ pairs for which there exists $p_{\hat{X}_{D,P}|Z}$ such that $\mathbb{E}[\Delta(X, \hat{X}_{D,P})] \leq D$ and $d(p_X, p_{\hat{X}_{D,P}}) \leq P$. Intuitively, $\Omega(p_{Z|X})$ is the set of all possible distortion-perception constraints that can be met based on representation $Z$. If $\Omega(p_{Z|X}) = \Omega(R)$ for some representation $Z$ with $I(X; Z) = R$, then $Z$ has the maximal achievable distortion-perception region in the sense that $\Omega(p_{Z'|X}) \subseteq \Omega(p_{Z|X})$ for any $Z'$ with $I(X; Z') \leq R$. In the supplementary material we establish mild regularity conditions for which the existence of such $Z$ is equivalent to the aforementioned desired property $A(\Omega(R)) = 0$. We shall show that that this ideal scenario actually arises in the Gaussian case and an approximate version can be found more broadly.

**Theorem 3.** *Let $X \sim \mathcal{N}(\mu_X, \sigma_X^2)$ be a scalar Gaussian source and assume MSE and $W_2^2(\cdot, \cdot)$ losses. Let $\Theta$ be any non-empty set of $(D, P)$ pairs. Then*

$$A(\Theta) = 0. \tag{7}$$

*Moreover, for any representation $Z$ jointly Gaussian with $X$ such that*

$$I(X; Z) = \sup_{(D,P) \in \Theta} R(D, P), \tag{8}$$

*we have*

$$\Theta \subseteq \Omega(p_{Z|X}) = \Omega(I(X; Z)). \tag{9}$$

Next we consider a general source $X \sim p_X$ and characterize the achievable distortion-perception region for an arbitrary representation $Z$ under MSE loss. We then provide some evidence indicating that every reconstruction $\hat{X}_{D,P}$ achieving some point $(D, P)$ on the distortion-perception tradeoff for a given $R$ likely has the property $\Omega(p_{\hat{X}_{D,P}|X}) \approx \Omega(R)$.

**Theorem 4** (Approximate universality for general sources). *Assume MSE loss and any perception measure $d(\cdot, \cdot)$. Let $Z$ be any arbitrary representation of $X$. Then*

$$\Omega(p_{Z|X}) \subseteq \left\{ (D, P) : D \geq \mathbb{E}[\|X - \tilde{X}\|^2] + \inf_{p_{\hat{X}}:d(p_X,p_{\hat{X}})\leq P} W_2^2(p_{\tilde{X}}, p_{\hat{X}}) \right\} \subseteq \mathrm{cl}(\Omega(p_{Z|X})),$$

*where $\tilde{X} = \mathbb{E}[X|Z]$ is the reconstruction minimizing squared error distortion with $X$ under the representation $Z$ and $cl(\cdot)$ denotes set closure. In particular, the two extreme points $(D^{(a)}, P^{(a)}) = (\mathbb{E}[\|X - \tilde{X}\|^2], d(p_X, p_{\tilde{X}}))$ and $(D^{(b)}, P^{(b)}) = (\mathbb{E}[\|X - \tilde{X}\|^2] + W_2^2(p_{\tilde{X}}, p_X), 0)$ are contained in $cl(\Omega(p_{Z|X}))$.*

To gain a better understanding, let $Z$ be an optimal reconstruction $\hat{X}_{D,P}$ associated with some point $(D, P)$ on the distortion-perception tradeoff for a given $R$, i.e., $I(X; \hat{X}_{D,P}) = R(D, P) = R$ (assuming that $D$ and/or $P$ cannot be decreased without violating $R(D, P) = R$), $\mathbb{E}[\|X - \hat{X}_{D,P}\|^2] = D$, $d(p_X, p_{\hat{X}_{D,P}}) = P$. We assume for simplicity that $\hat{X}_{D,P}$ exists for every $(D, P)$ on the tradeoff. Such $(D, P)$ is on the boundary of $\mathrm{cl}(\Omega(p_{\hat{X}_{D,P}|X}))$. Theorem 4 indicates that $\mathrm{cl}(\Omega(p_{\hat{X}_{D,P}|X}))$ contains two extreme points: the upper-left $(D^{(a)}, P^{(a)})$ and the lower-right $(D^{(b)}, P^{(b)})$. Under the assumption that $d(\cdot, \cdot)$ is convex in its second argument, $\mathrm{cl}(\Omega(p_{\hat{X}_{D,P}|X}))$ is a convex region containing the aforementioned points.

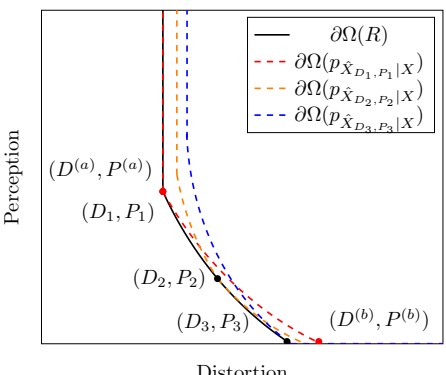

Figure 2: Approximate universality for a general source. Illustrated are boundaries of achievable distortion-perception regions for three representations: minimal distortion $(D_1, P_1)$ for $R(D_1, P_1) = R(D_1, \infty)$, midpoint $(D_2, P_2)$, and perfect perceptual quality $(D_3, P_3)$ where $P_3 = 0$. We have $\Omega(p_{\hat{X}_{D_i, P_i}|X}) \approx \Omega(R)$, especially when $R$ is small. The extreme points $(D^{(a)}, P^{(a)})$ and $(D^{(b)}, P^{(b)})$ for $\hat{X}_{D_1, P_1}$ are shown. $(D^{(a)}, P^{(a)})$ coincides with $(D_1, P_1)$.

Figure 2 illustrates $\Omega(R)$ and $\Omega(p_{\hat{X}_{D,P}|X})$ for several different choices of $(D, P)$. When $R = 0$, $\Omega(p_{\hat{X}_{D,P}|X}) = \Omega(R)$ for any such $\hat{X}_{D,P}$. So we have $\Omega(p_{\hat{X}_{D,P}|X}) \approx \Omega(R)$ in the low-rate regime where the tension between distortion and perception is most visible. More general quantitative results are provided in the supplementary. Let $\sigma_X^2 = \mathbb{E}[\|X - \mathbb{E}[X]\|^2]$. If $\hat{X}_{D_1, P_1}$ is chosen to be the optimal reconstruction in the conventional rate-distortion sense associated with point $(D_1, P_1)$, then the upper-left extreme points of $\Omega(p_{\hat{X}_{D_1, P_1}|X})$ and $\Omega(R)$ coincide (i.e., $(D^{(a)}, P^{(a)}) = (D_1, P_1)$) and the lower-right extreme points of $\Omega(p_{\hat{X}_{D_1, P_1}|X})$ and $\Omega(R)$ (i.e., $(D^{(b)}, 0)$ and $(D_3, 0)$ with $R(D_3, 0) = R(D_1, \infty)$) must be close to each other in the sense that

$$\frac{1}{2}\sigma_X^2 \geq D^{(b)} - D_3 \overset{D_1 \approx 0 \text{ or } \sigma_X^2}{\approx} 0, \qquad 2 \geq \frac{D^{(b)}}{D_3} \overset{D_1 \approx \sigma_X^2}{\approx} 1, \tag{10}$$

which suggests that $\Omega(p_{\hat{X}_{D_1, P_1}|X})$ is not much smaller than $\Omega(R)$. Moreover, in this case we have

$$D^{(b)} \leq 2\mathbb{E}[\|X - \tilde{X}\|^2] \leq 2D_1, \tag{11}$$

which implies that $(2D_1, 0)$ is dominated by extreme point $(D^{(b)}, 0)$ and consequently must be contained in $\mathrm{cl}(\Omega(p_{\hat{X}_{D_1, P_1}|X}))$. Therefore, the optimal representation in the conventional rate-distortion sense can be leveraged to meet any perception constraint with no more than a two-fold increase in distortion. As a corollary, one recovers Theorem 2 in Blau & Michaeli ($R(2D, 0) \leq$

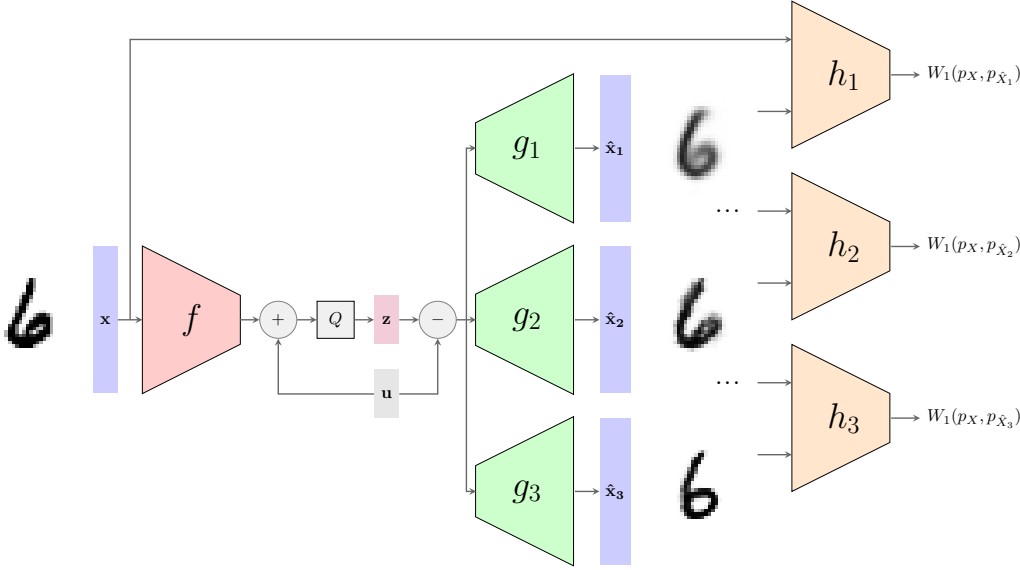

Figure 3: An illustration of the experimental setup for the universal model. A single encoder $f$ is trained for an initial perception-distortion tradeoff and has its weights frozen. Subsequently many other decoders $\{g_i\}$ are optimized for different tradeoff points using the representations $z$ produced by $f$. The sender and receiver have access to a shared source of randomness $u$ for universal quantization [30, 42]. $Q$ denotes the quantizer. Separate critic networks $\{h_i\}$ are trained along with each decoder to promote perceptual quality. In this figure, the top decoder places most weight on distortion loss whereas the bottom decoder places most weight on perceptual loss. This has the effect of reducing the blurriness, but comes at the cost of a less faithful reconstruction of the original (in extreme cases even changing the identity of the digit). The perception losses $W_1(p_X, p_{\hat{X}_i})$ are estimated using the critics $\{h_i\}$ by replacing the expectations in Equation (14) with samples from the test set.

$R(D, \infty)$). Hence, the numerical connection between $R(2D, 0)$ and $R(D, \infty)$ is a manifestation of the existence of approximately $\Omega(I(X; Z))$-universal representations $Z$. These analyses motivate the study of practical constructions for which we seek to achieve multiple $(D, P)$ pairs with a single encoder.

### 3.3 Successive Refinement

Up until now, we have established the notion of distortion-perception universality for a given rate. We can paint a more complete picture by extending this universality along the rate axis as well, known classically as successive refinement [9] when restricted to the rate-distortion function. Informally, given two sets of $(D, P)$ pairs $\Theta_1$ and $\Theta_2$, we say that rate pair $(R_1, R_2)$ is (operationally) *rate-distortion-perception refinable* if there exists a base encoder optimal for $R(\Theta_1)$ which, when combined with a second refining encoder, is also optimal for $R(\Theta_2)$. In other words, bits are transmitted in two stages and each stage achieves optimal rate-distortion-perception performance. This nice property is not true of general distributions but we show in supplementary section A.3 that it holds in the asypmtotic Gaussian case, thus generalizing Theorem 3. Nonetheless, building on [18] we prove an approximate refinability property of general distributions and in section B.3 provide experimental results demonstrating approximate refinability on image compression using deep learning.

## 4 Experimental Results

The rate-distortion-perception tradeoff was observed as a result of applying GAN regularization within deep-learning based image compression [5, 33]. Therein, an entire end-to-end model is trained for each desired setting over rate, distortion, and perception. In practice it is undesirable to develop an entire system from scratch for each objective and we would like to reuse trained networks with

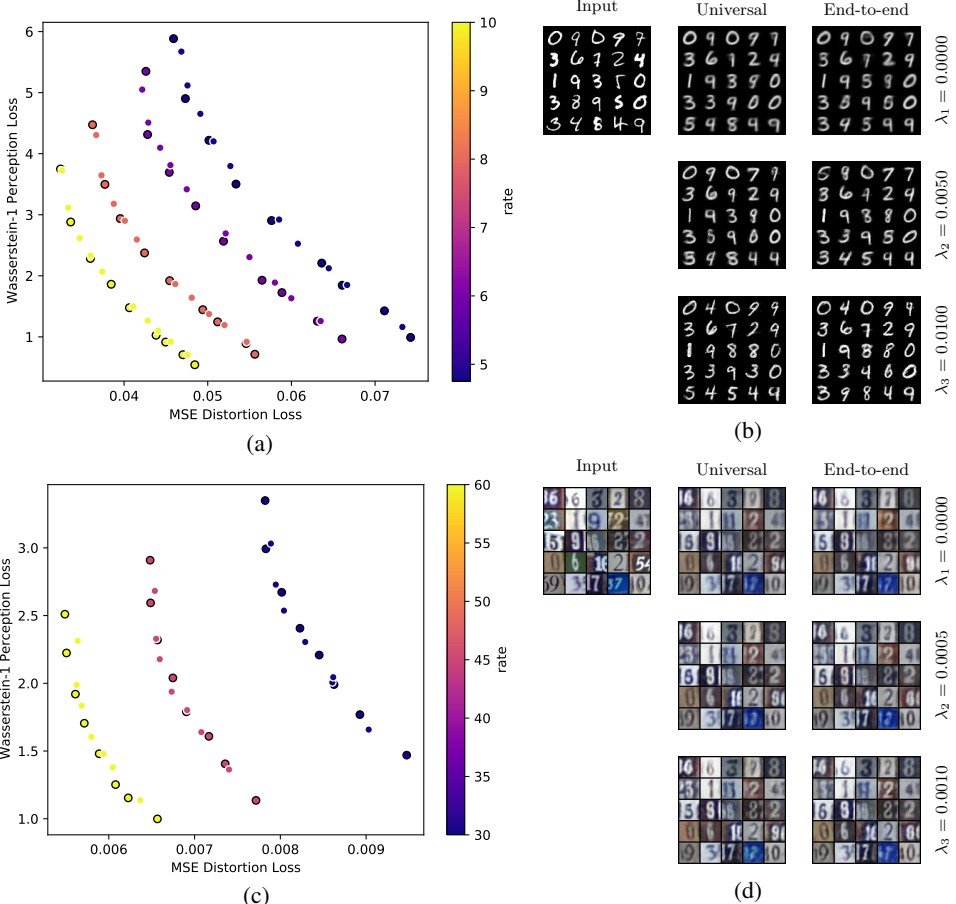

Figure 4: (a) (c) Rate-distortion-perception tradeoffs along various rates. Points with black outline are losses reported for the end-to-end encoder-decoder pairs trained jointly for a particular perception-distortion objective. Other points are the losses for universal models, in which decoders are trained over a frozen encoder optimized for small $P$ (MNIST: $\lambda = 0.015$, SVHN: $\lambda = 0.002$). Universal model performance is very close to performance of end-to-end models across all tradeoffs $\{\lambda_i\}$. (b) (d) Outputs of selected models (MNIST: $R = 6$, SVHN: $R = 60$). As the emphasis on perception loss $\lambda_i$ increases, the outputs become sharper. The visual quality of both the end-to-end and universal models are on average comparable for each $\lambda_i$. More experiment details are given in the supplementary.

frozen weights if possible. It is of interest to assess the distortion and perception penalties incurred by such model reusage, most naturally in the scenario of fixing a pre-trained encoder.

Concretely, we refer to models where the encoder and decoder are trained jointly for an objective as *end-to-end* models, and models for which some encoder is fixed in advance as (approximately) *universal* models. The encoders used within the universal models are borrowed from the end-to-end models, and the choice of which to use will be discussed later in this section. Within the same dataset, universal models and end-to-end models using the same hyperparameter settings differ only in the trainability of the encoder.

## 4.1 Setup and Training

The architecture we use is a stochastic autoencoder with GAN regualarization, wherein a single model consists of an encoder $f$, a decoder $g$, and a critic $h$. Details about the networks can be found in the supplementary; here, we summarize first the elements relevant to facilitating compression then the training procedure. Let $x$ be an input image. The final layer of the encoder consists of a tanh activation to produce a symbol $f(x) \in [-1, 1]^d$, with the intent to divide this into $L$-level intervals of uniform length $2/(L-1)$ across $d$ dimensions for some $L$. This gives an upper bound

of $d \log(L)$ for the model rate, and it was found to be only slightly suboptimal by Agustsson et al. [1]; we found the estimate to be off by at most 6% on MNIST. To achieve high perceptual quality through generative modelling, stochasticity is necessary[1] [33]. In accordance with the shared randomness assumption within the ouRDPF, our main experimental results use the universal/dithered quantization[2] [12, 29, 30, 42] scheme where the sender and receiver both have access to a sample $u \sim U[-1/(L-1), +1/(L-1)]^d$. The sender computes

$$z = \text{Quantize}(f(x) + u) \tag{12}$$

and gives $z$ to the receiver. The receiver then reconstructs the image by feeding $z - u$ to the decoder. The soft gradient estimator of [21] is used to backpropagate through the quantizer. Compared to alternate schemes where noise is added only at the decoder, this scheme has the advantage of reducing the quantization error by centering it around $f(x)$ and can be emulated on the agreement of a random seed. Since this is not always possible in practice, the results for a more restrictive quantization scheme where the sender and receiver do not have access to common randomness are included in Figure 5 in the supplementary.

The rest of the design follows closely the design of Blau & Michaeli [5]. We first produce the end-to-end models, in which $f, g$ and $h$ are all trainable. We use MSE loss for the distortion metric and estimate the Wasserstein-1 perception metric. The loss function is given by

$$\mathcal{L} = \mathbb{E}[\|X - \hat{X}\|^2] + \lambda W_1(p_X, p_{\hat{X}}), \tag{13}$$

where $p_{\hat{X}}$ is the reconstruction distribution induced by passing $X$ through $f$, transmitting the representations via (12) then subtracting the noise and decoding through $g$. The particular tradeoff point achieved by the model is controlled by the weight $\lambda$. Kanotorovich-Rubinstein duality allows us to write the Wasserstein-1 distance as

$$W_1(p_X, p_{\hat{X}}) = \max_{h \in \mathcal{F}} \mathbb{E}[h(X)] - \mathbb{E}[h(\hat{X})], \tag{14}$$

which expresses the objective as a min-max problem and allows us to treat it using GANs. Here, $\mathcal{F}$ is the set of all bounded 1-Lipschitz functions. In practice, this class is limited by the discriminator architecture and the Lipschitz condition is approximated with a gradient penalty [13] term. Optimization alternates between minimizing over $f, g$ with $h$ fixed and maximizing over $h$ with $f, g$ fixed. In essence, $g$ is trained to produce reconstructions that are simultaneously low distortion and high perception, so it acts as both a decoder and a generator. The reported perception loss is estimated using Equation (14) through test set samples. Figure 3 provides an overview of the entire scheme.

After the end-to-end models are trained, their encoders can be lent to construct universal models. The parameters of $f$ are frozen we introduce a new decoder $g_1$ and critic $h_1$ trained to minimize

$$\mathcal{L}_1 = \mathbb{E}[\|X - \hat{X}_1\|^2] + \lambda_1 W_1(p_X, p_{\hat{X}_1}),$$

where $\lambda_1$ is another tradeoff parameter and $p_{\hat{X}_1}$ is the new reconstruction distribution. The weights of $g_1$ are initialized from random while the weights of $h_1$ are initialized from $h$. This was done for stability and faster convergence but in practice, we found that initializing from random performed just as well given sufficient iterations. The rest of the training procedure follows that of the first stage. This second stage is repeated over many different parameters $\{\lambda_i\}$ to generate a tradeoff curve. Further model and experimental details can be found in the supplementary material.

## 4.2 Results

Figure 4 shows rate-distortion-perception curves at multiple rates on MNIST and SVHN, obtained by varying $\lambda$ from 0 to a selected upper bound for which training with the given hyperparameters remained stable. Note that the rate for each individual curve is fixed through using the same quantizer across all models. As the rate is increased by introducing better quantizers, optimizing for distortion loss has the side effect of reducing perception loss. The rates are thus chosen to be low as the tension between distortion and perception is most visible then. The points outlined in black are losses for end-to-end models and the other points correspond to the universal models sharing an encoder trained

---

[1]This prevents us from passing noiseless quantized representations to the decoder.
[2]The use of the word "universal" here is unrelated to our notion of "universality".

from the end-to-end models. As can be seen, the universal models are able to achieve a tradeoff which is very close to the end-to-end models (with outputs that are visually comparable) despite operating with a fixed encoder.

For any fixed rate, decreasing the perception loss $P$ induces outputs which are less blurry, at the cost of a reconstruction which is less faithful to the original input. This is especially evident at very low rates in which the compression system appears to act as a generative model. However, our experiments indicate that an encoder trained for small $P$ can also be used to produce a low-distortion reconstruction by training a new decoder. Conversely, training a decoder to produce reconstructions with high perceptual quality on top of an encoder trained only for distortion loss is also possible as the decoder is sufficiently expressive to act purely as a generative model.

## 5 Discussion

**Limitations**. One limitation of these experiments is that we can slightly reduce the distortion loss by using deterministic nearest neighbour quantization rather than universal quantization, but there would no longer be stochasticity to train the generative model. A comparison of quantization schemes for the case of $\lambda = 0$ can be found in Table 1 of the supplementary. It may be beneficial to employ more sophisticated quantization schemes and explore losses beyond MSE as well.

**Potential Negative Societal Impacts**. The goal of our work is to advance perceptually-driven lossy compression, which conflicts with optimizing for distortion. We presume that this will be harmless in most multimedia applications but where reconstructions are used for classification or anomaly detection this may cause problems. For example, a low-rate face reconstruction deblurred by a GAN may lead to false identity recognition.

## 6 Conclusion

The use of deep generative models in data compression has highlighted the tradeoff between optimizing for low distortion and high perceptual quality. Previous works have designed end-to-end systems in order to achieve points across this tradeoff. Our results suggest that this may not be necessary, in that fixing a good representation map and varying only the decoder is sufficient for image compression in practice. We have also established a theoretical framework to study this scheme and characterized its limits, giving bounds for the case of specific distributions and loss functions. Future work includes evaluating the scheme on more diverse architectures, as well as employing the scheme to high-resolution images and videos.

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
