# OpenReview forum: "Universal Rate-Distortion-Perception Representations for Lossy Compression"
_NeurIPS.cc/2021/Conference — NeurIPS 2021 Poster_

### Official Review · Reviewer_ZPE7 · 2021-07-09

**Rating:** 7
**Confidence:** 3

**Summary:**

The authors present a theoretical contribution to the rate-distortion-perception framework for lossy compression recently proposed by Blau and Michaeli. The authors show that it is possible to create universal representations by fixing an encoder (and hence rate) and change decoders to achieve multiple distortion-perception tradeoffs, with minimal penalty with respect to variable encoders.

**Ethics Review Area:**

["I don’t know"]

**Limitations And Societal Impact:**

Societal impact and limitations have been adequately discussed.

**Main Review:**

Overall, the paper is well written and rigorously proves the authors' claim. Exploring the new framework of rate-distortion-perception theory is an interesting topic and important to advance the results on lossy compression, in light of recent results on perceptual quality achieved by models like GANs. To the best of my knowledge, the presented result is novel and might have interesting implications in the design of new image compression schemes. The paper is a theoretical contribution but the authors also show some experimental results on the MNIST and SVNH datasets. While these datasets are too simple to extrapolate the results to more realistic image compression, they are adequate at showing the theoretical points discussed by the authors.

**Time Spent Reviewing:**

2

---

> ### Author Response · Authors · 2021-08-09
> **Response to Reviewer ZPE7**
>
> Thank you for your feedback. We are happy to hear that you think this work has the potential to be impactful in the future.

---

### Official Review · Reviewer_1XEU · 2021-07-15

**Rating:** 8
**Confidence:** 4

**Summary:**

The paper considers the rate-distoriton-perception trade-off setting, and studies what happens if we fix an encoder and train different decoders for different (D, P) targets. This is studied both theoretically and empirically. It is shown that for Gaussian sources, the rate penalty incurred by using a single encoder for multiple (D, P) target is zero. Empirically, it is shown that on MNIST and SVHN, jointly training decoders and encoders is only marginally better in terms of D-P than when using a frozen encoder.

**Limitations And Societal Impact:**

The paper contains an adequate discussion of social impacts and limitations.

**Main Review:**

The paper is well written, studies an interesting setting, and contains solid results. The math is at parts a bit dense and sometimes, intuition could have been provided before the formalism, but I was able to follow everything in the main text. I had more trouble following the proofs in the supplementary in detail, and was not able to verify them.

The only question I had was regarding L273: why are the critics initialized from the end-to-end model? Is this required?

The following are minor comments. Most are cosmetic, so feel free to ignore, as I also do like the clean notation that is used.
- L32: Could use a "\emph" on "distributions" for people new to this.
- L50: compare -> compared
- L54: It's a bit sad that Section A.3 on successive refinement is only mentioned very briefly in the main text. Possibly, a summary of the empirical results could be added to the main text, as it seems this setting may be interesting for practical full-resolution compression.
- L119: uRDP was never introduced (although it is clear what it means)
- L136: I find the overloading of $R^\star$ depending on whether the argument is a single (D, P) pair or a set of pairs slightly confusing. Possibly, the latter could be $R^\star_u$, at the price of more symbols littered around.
- L138 (Def 3): Here, it would have been instructive to start by saying that Z will be the representation of X.
- L138 (Def 3): I would have put the "for each (D, P) $\in \Theta$" before the equation, close to $p_{\hat X_{D,P}}$, which uses D, P.
- L154: The symbol $A$ is not very memorable. Possibly something like $R_\mathcal{E}$ would be more suited, or otherwise highlighting the "all" on L155 (where the A presumably originates from?) would be memorable.
- L167: $\Omega$ is also overloaded, this time depending on whether a distribution or a rate is used.
- Eq 12: L is already used for levels, maybe $\mathcal{L}$ would be more common?

**Time Spent Reviewing:**

4

---

> ### Author Response · Authors · 2021-08-09
> **Response to Reviewer 1XEU**
>
> We thank the reviewer for the detailed suggestions, which are greatly appreciated. We will incorporate this feedback during our revisions to make the proofs more readable. We are also glad that there is interest in successive refinement.
>
> Regarding L273: The original intention was to speed up training, but after running an experiment for a sufficiently long time as we have done in the paper we believe that this initialization is not necessary. We have performed a quick comparison on a subset of the experiments run without the critic initialization on the MNIST dataset and found the results to be essentially the same as the reported results; we will add a comment about this in the paper.

---

### Official Review · Reviewer_nYfy · 2021-07-16

**Rating:** 7
**Confidence:** 3

**Summary:**

The paper suggests that previous end-to-end compression system do not need to train different encoders to achieve points across rate-distortion-perception curves but just fixes a good representation map and varying only the decoder is sufficient. The paper also establishes a theoretical framework to support the conclusion.

**Limitations And Societal Impact:**

Authors have discussed the limitations and negative societal impact.

**Main Review:**

Originality: Good. The paper proposes a framework to achieve different distortion-perception tradeoff by training different decoder with a fixed encoder to and provide corresponding theoretical analysis, which ensures novelty and provides theoretical base for future study.

Quality: Good. The submission is technically sound in my opinion and the advantages and limitations of this work are discussed carefully and honestly. The rate-distortion-perception tradeoff is well analyzed and serves as theoretical base for the practical framework.

Clarity: Good. The submission is written with sufficient clear definitions and formulas. And the organizations are well-designed.

Significance: Kind of doubtful. Considering different distortion-perception trade-off is not very common for compression, and usually this trade-off is fixed depended on the specific task. Therefore, the proposed framework does provides a solution to decreasing training resource facing this trade-off but I doubt that whether it is really a useful tool. However, the paper proves its mention well enough, both theoretically and practically, which is brilliant.

**Time Spent Reviewing:**

3 hours

---

> ### Author Response · Authors · 2021-08-09
> **Response to Reviewer nYfy**
>
> Thank you for the positive comments. With regards to the significance of our work, we hope to also inspire further works which are built on top of existing compression schemes or popular deep learning architectures optimized for distortion encoding to operate with perceptual quality in mind.

---

### Official Review · Reviewer_Wozp · 2021-07-16

**Rating:** 8
**Confidence:** 3

**Summary:**

The rate-distortion-perception (RDP) function[5,23] is a generalization of the classical rate-distortion (RD) trade-off in lossy compression to also measure realism, which establishes a theoretical footing for generative image/video compression, a topic of active research[1,11,17,25].

This paper studies the RDP function under the constraint of a shared encoder for multiple D-P points, both from the theoretical side and experimentally. This is of significant practical interest, as detailed below.

**Ethical Concerns:**

None.

**Limitations And Societal Impact:**

The limitations and societal impacts were adequately addressed by the authors.

**Main Review:**

So far, it has been assumed that an encoder/decoder system will be need to be optimized for a particular trade-off between distortion and realism (perception). This is not ideal, as at encode time we may not know what we want to use the reconstructed images for. If they end up in the courtroom, we would not want the reconstructions to contain synthesized detail (despite having high realism) --- if they are displayed for entertainment purposes, we may not want them to be blurry (despite having lower distortion).

The topic of this paper is to study universal representations for the RDP trade-off, such that a single encoding could be decoded in multiple ways depending on the the desired D-P trade-off.
This is of significant practical interest, as this simplifies the problem at hand.

The paper makes the following contributions:
* It builds a convincing theory of the RDP trade-off for universal representations. This includes:
- Characterizing their trade-off operationally and informationally ( Def. 2&3 + Thm. 2.)
- Showing universality is essentially "free" for gaussian sources + MSE distortion + Wasserstein distance (Thm. 3.)
- Shows in Thm. 4 is an "approximate universality" for the general case of arbitrary distributions ( with MSE distortion).
- Demonstrates experimentally on toy datasets (MNIST, SVHN) that the D-P trade-off for various bitrates can be navigated from a single representation without a significant cost in performance.

A limitation of the work is that the experiments only considers low resolution images, but I consider that appropriate given the theoretical focus of the work.

Given the both the theoretical insights this paper provides, as well as their potential practical value,
I recommend this paper for acceptance.

Final update: Having read the other reviews, I am confident in keeping my score.

**Time Spent Reviewing:**

3

---

> ### Author Response · Authors · 2021-08-09
> **Response to Reviewer Wozp**
>
> Thanks for the positive comments. We are glad that there is continued interest in the development of a theoretical framework to complement the empirical success of perceptually-driven lossy compression.

---

### Decision · Program_Chairs · 2021-09-27

**Decision:**

Accept (Poster)

**Comment:**

The paper provides an interesting contribution of theoretical nature, that essentially shows that one can work with a single encoder but vary the decoder to achieve different rate-distortion-perception trades-off in (image) compression. The work is likely to stimulate interesting discussion at NeurIPS, and provides useful insights towards future research in lossy compression.